# Parasagittal dural volume correlates with cerebrospinal fluid volume and developmental delay in children with autism spectrum disorder
Nivedita Agarwal [1] ✉, Giulia Frigerio [1], Gloria Rizzato [1], Tommaso Ciceri [2,3], Elisa Mani[4], Fabiola Lanteri[4], Massimo Molteni[4], Roxana O. Carare [5,6], Letizia Losa [1] & Denis Peruzzo [2]

## Abstract

**Background** The parasagittal dura, a tissue that lines the walls of the superior sagittal sinus, acts as an active site for immune-surveillance, promotes the reabsorption of cerebrospinal fluid, and facilitates the removal of metabolic waste products from the brain. Cerebrospinal fluid is important for the distribution of growth factors that signal immature neurons to proliferate and migrate. Autism spectrum disorder is characterized by altered cerebrospinal fluid dynamics.

**Methods** In this retrospective study, we investigated potential correlations between parasagittal dura volume, brain structure volumes, and clinical severity scales in young children with autism spectrum disorder. We employed a semi-supervised two step pipeline to extract parasagittal dura volume from 3D-T2 Fluid Attenuated Inversion Recovery sequences, based on U-Net followed by manual refinement of the extracted parasagittal dura masks.

**Results** Here we show that the parasagittal dura volume does not change with age but is significantly correlated with cerebrospinal fluid (p-value = 0.002), extra-axial cerebrospinal fluid volume (p-value = 0.0003) and severity of developmental delay (p-value = 0.024).

**Conclusions** These findings suggest that autism spectrum disorder children with severe developmental delay may have a maldeveloped parasagittal dura that potentially perturbs cerebrospinal fluid dynamics.

## Plain language summary

Cerebrospinal fluid (CSF) is produced in the brain. It is a medium of transport for neural growth factors and waste products. CSF is drained out of the brain through multiple pathways, one of them being the recently identified parasagittal dura (PSD) which also plays a role in the immune system within the brain. We estimated the PSD volume in children with autism spectrum disorder (ASD) and found the volume was associated with the amount of CSF in the brain. We also found that the PSD volume is smaller in children who have severe forms of developmental delay. Our findings suggest problems in the development of the PSD could have in impact on brain development and waste removal in children with ASD. More research in this area could enable a better understanding of the underlying causes of ASD.

The parasagittal dura (PSD) is a parasinus tissue located along the exterior walls of the superior sagittal sinus[1]. The PSD hosts meningeal lymphatic channels, stromal elements, immune cells, and arachnoid granulations[2–5]. Recent experimental studies demonstrate that this dura-arachnoid tissue serves multiple roles: acts as a conduit for the flow of cerebrospinal fluid (CSF) towards meningeal lymphatics (MLs), facilitates the elimination of metabolic waste from the brain and plays a pivotal role in brain immune-surveillance[6–10]. PSD contains diverse immune cell subsets actively monitoring for cerebral antigens that find their way into peripheral lymph nodes[11,12].

There are few studies that have employed magnetic resonance imaging (MRI) to quantify the volume of PSD in adults[13–15]. PSD volumes increase during lifespan which is likely a compensatory response to age-related impairment of the lymphatic drainage and MLs[13,14,16]. In patients with Alzheimer's disease, PSD volume was directly correlated with a greater load of amyloid beta deposition in the brain

[1]Diagnostic Imaging and Neuroradiology Unit, IRCCS Scientific Institute E. Medea, Bosisio Parini, Lecco LC, Italy. [2]Neuroimaging Unit, IRCCS Scientific Institute E. Medea, Bosisio Parini, Lecco LC, Italy. [3]Department of Information Engineering, University of Padua, Padua, Italy. [4]Child Psychopathology Unit, IRCCS Scientific Institute E. Medea, Bosisio Parini, Lecco LC, Italy. [5]Faculty of Medicine, University of Southampton, Southampton, UK. [6]University of Medicine, Pharmacy, Science, and Technology, Targu-Mures, Romania. ✉e-mail: nivedita.agarwal@lanostrafamiglia.it

parenchyma, suggesting that a hypertrophic PSD reflects altered dynamics in neurofluids and poor waste clearance[17].

Autism Spectrum Disorder (ASD) is a complex neurodevelopmental disorder characterized by heterogeneous manifestations of symptoms. These include stereotypical behaviors and social and communication skill deficits[18]. Epidemiologic studies suggest that the prevalence of ASD is increasing worldwide estimated at 27.6 per 1000 children[19]. The increase in prevalence is likely a combination of enhanced diagnostic criteria but also the presence of more recently discovered epigenetic and multiple environmental factors. The etiology of ASD remains largely elusive with both genetic and environmental factors being variably involved in the expression of ASD phenotype[18,20,21].

Some evidence suggests that CSF dynamics are disrupted in ASD, potentially due to an imbalance between CSF production and absorption[22,23]. Furthermore, several studies have suggested that immunological dysregulation in children with ASD initiates a subtle neuroinflammatory process that hinders typical development of the central nervous system[24,25].

MRI is a non-invasive tool to study the anatomy, biochemistry, and function of the brain. While the diagnosis of ASD is based mostly on clinical scales, an MRI is usually requested to rule out structural or organic etiologies of cognitive dysfunction[26]. To date, no studies have evaluated PSD in the developing brain. Our objective was to delineate PSD within our in-patient ASD cohort and explore potential correlations between PSD volume, brain tissue volumes, and clinical severity scales in ASD by utilizing a semi-automatic segmentation pipeline including a convolutional neural network and manual refinement. Our results suggest that PSD volume does not change with age but is significantly correlated with CSF, extra-axial cerebrospinal fluid volume, and severity of developmental delay in patients with ASD.

## Methods

### Ethical approval

This retrospective study was approved by the IRCCS Eugenio Medea Institutional Review Board (Protocol No. 1022) and written informed consent was obtained from all legal representatives (parents or legal guardians) of the children.

### Study participants

Children clinically diagnosed with ASD were selected for this study. The diagnosis was conducted by a multidisciplinary team at the Child Psychopathology Unit of the Scientific Institute IRCSS E. Medea (Bosisio Parini, Italy), according to DSM-5 criteria (American Psychiatric Association, 2013. Diagnostic and Statistical Manual of Mental Disorders. Fifth edition. Washington, DC: American Psychiatric Association) and regardless of the presence of global developmental delay or intellectual disability. The diagnostic instruments employed included the Autism Diagnostic Interview–Revised (ADI-R)[27] administered to parents and the Autism Diagnostic Observation Schedule-second edition (ADOS-2)[28] conducted with the child. The Calibrated Severity Score (CSS) was employed as a metric for assessing the severity of autistic symptoms[29,30]. The scale ranges from 1 to 10, classifying severity into three categories: 1–3 for non-spectrum, 4–5 for autism spectrum disorder, and 6–10 for autism. IQ was assessed using either the Wechsler Intelligence Scale for Children (WISC-IV)[31] or the Wechsler Preschool and Primary Scale of Intelligence-III (WPPSI-III)[31] selecting the test based on the child's age and cognitive-linguistic abilities. For children unable to complete these tests due to lack of cooperation, age, or absence/difficulty with language, we conducted a psychomotor development assessment using the Griffiths Mental Development Scales (cGMDS-ER)[32]. IQ scores were further grouped into four classes: normal (>70), mild (50–70), moderate (35–49), and severe (20–34). This classification was preferred over the use of a continuous variable because we believe that global functioning is a variable that correlates better with neuroradiological data than small numerical variations within the functioning class. As part of the clinical diagnostic process, all children underwent brain MRI examinations, as well as etiologic instrumental investigations, such as electroencephalograms and genetic tests, between January 2022 and March 2023. The initial clinical sample consisted of a total of 67 patients with a diagnosis of ASD. The following criteria led to the exclusion of patients from the study: (1) age less than 2 or greater than 8 years and; (2) reduced MRI quality. As a result, this retrospective study included a total of 56 children.

### MRI acquisition protocol

All our participants were sedated with continuous intravenous infusion of propofol. MRI data were acquired on a 3T scanner (Achieva dStream; Philips Medical Systems) with a 32-channel head coil at the Diagnostic Imaging and Neuroradiology Unit of the Institute. The MRI protocol included two anatomical sequences: (a) 3D-T1 weighted (3D-T1w): sagittal scanning plane; repetition time (TR) = 8,3 ms; echo time (TE) = 3,9 ms; echo train length (ETL) = 256; flip angle = 8°; 1 average; $1 \times 1 \times 1$ mm$^3$ voxel size. Acquisition time: 5 min and 38 s; (b) 3D-T2 Fluid Attenuated Inversion Recovery (3D-FLAIR): sagittal scanning plane; TR = 4800 ms; TE = 298 ms; inversion time = 1650 ms; ETL = 167; flip angle = 90°; 2 averages; $1 \times 1 \times 1$ mm$^3$ voxel size. Acquisition time: 6 min.

### MRI volumetric assessment

3D-T1w images were processed using an ad-hoc pipeline developed in-house which briefly consists in the following steps: (1) brain extraction from the acquired images combining multiple tools [BET, ROBEX, ANTS][33–35], (2) bias field intensity artifacts correction using the N4BC algorithm[36], (3) rigid registration to MNI space[37], and (4) segmentation of the main brain structures with Atropos using the PTBP (Pediatric Template of Brain Perfusion) priors[34]. From the processed 3D-T1w images the following volumes were derived for each child: ICV, CSF, WM, and cGM. The ea-CSF was derived from the CSF mask by manually removing the ventricles and the component below the anterior commissure – posterior commissure line (AC-PC line) (Fig. 1)[23].

### PSD segmentation and volumetric assessment

PSD segmentation was obtained from 3D-FLAIR images as they provide a larger contrast between the PSD and the CSF than the 3D-T1w images. Acquired images were processed using the N4 algorithm to remove any bias field intensity artifact.

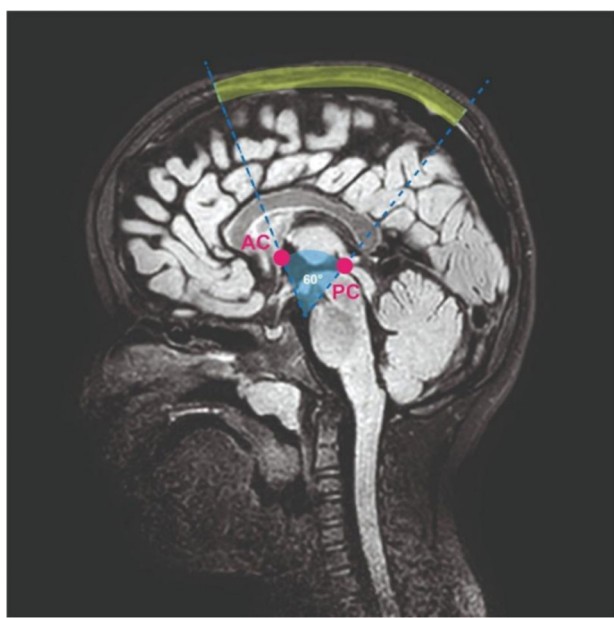

**Fig. 1 | PSD segmentation criteria on 3D-T2 Fluid Attenuation Inversion Recovery sequence.** AC anterior commissure and PC posterior commissure. The segmented PSD is colored yellow.

In this context, convolutional neural networks (CNNs), particularly the U-Net architecture and its variants, have become the state-of-the-art approach to perform an automatic and user-independent segmentation[38,39]. Thus, we developed an in-house 2D U-Net backbone-based architecture with the intent to facilitate the segmentation process[40]. Notably, the network was trained on an independent dataset including 10 healthy adults (32.6 ± 12.5 years) whose images were manually segmented by an expert neuroradiologist. Each participant dataset comprised at least 150 images for a total training set of 2250 coronal images. Validation was performed on 418 coronal images from 2 healthy adults and the test set comprised 1941 coronal images derived from 10 ASD children. This U-Net was then applied to our cohort of 56 children with ASD. All the resulting segmentations were manually refined by the neuroradiologist to correct for erroneous segmentations. The performance metrics between the U-Net-based automatic segmentation results and the manually corrected segmentations are presented in Supplementary Figs. S1 and S2. The U-Net architecture was employed given the constrained training dataset, as implemented in previous PSD segmentation studies[39].

The anterior and the posterior segments of PSD in the very young developing brain are either absent or very difficult to disentangle from the surrounding brain structures. Furthermore, the PSD aspect in the anterior and posterior segments is very different from the central one due to the relative inclination of the coronal plane with the PSD skeleton direction. As a consequence, we decided to restrict the PSD segmentation to its central components to enhance reproducibility. More precisely, we delineated a region of interest for the PSD segmentation by tracing an arc on the cranial circumference, subtended by a 60° angle passing through the anterior commissure-posterior commissure (AC-PC) landmarks (Fig. 1). Finally, the volume of the central component of the PSD was derived from the segmentation and used as a proxy of the whole PSD volume.

## Statistics

This study involved the presence of both continuous (e.g., brain structure volumes, age) and categorical variables (e.g., IQ classes, ADOS). The normal distribution of variables was verified with the Shapiro–Wilk test and compared using the student's $t$-test for independent samples. Correlations were determined by Kendall correlation tests. ANOVA test was used for assessing differences between groups in categorical variables. Data were analyzed using statistical analysis with R setting the significant threshold for the $p$-value to 0.05 or $p$-value of 0.01 in the case of Bonferroni correction. In the analysis comparing PSD volume with five distinct cerebral volumes (ICV, WM, cGM, CSF, and ea-CSF), the Bonferroni correction was utilized to tackle the issue of multiple comparisons in statistical testing. This correction involved setting the $p$-value threshold at 0.01 (calculated as 0.05 divided by the number of comparisons), ensuring a more stringent criterion for determining statistical significance in each individual comparison.

## Reporting summary

Further information on research design is available in the Nature Portfolio Reporting Summary linked to this article.

## Results
### Study participants

The study included 56 patients with confirmed ASD diagnoses that met the inclusion criteria defined in the Materials and Methods section. The characteristics of the study participants are provided in Table 1.

### Brain volumetrics and age

The average volumes of intracranial volume (ICV), cortical gray matter (cGM), white matter (WM), CSF, extra-axial CSF (ea-CSF), and PSD are reported in Table 2. A significant age-related increase in ICV ($R = 0.28$, $p$-value = 0.00027), WM ($R = 0.34$, $p$-value = 0.0002), and cGM ($R = 0.18$, $p$-value = 0.049) was observed. In contrast, no significant correlations were found with, CSF, ea-CSF, and PSD with age, as depicted in Fig. 2.

**Table 1 | Demographic and clinical characteristics of patients**

|  | Male ($N$ = 48) | Female ($N$ = 8) |
|---|---|---|
| **Age (years)** | | |
| Mean ± SD | 4.5 ± 1.5 | 3.9 ± 1.5 |
| **ADOS-2 CSS[a]** | Number of patients | Number of patients |
| • ASD | 12 | 1 |
| • Autism | 35 | 7 |
| • N/A[b] | 1 | 0 |
| **IQ class[c]** | Number of patients | Number of patients |
| • Normal | 12 | 0 |
| • Mild | 11 | 3 |
| • Moderate | 9 | 0 |
| • Severe | 15 | 2 |
| N/A[b] | 1 | 3 |

[a]Autism Diagnostic Observatory Schedule–Clinical Severity Score (ADOS-CSS).
[b]N/A = not available.
[c]Intelligent Quotient class.

**Table 2 | Mean volumes of all brain structures**

| Brain Structure | Volumes [cm³] (Mean ± SD) | |
|---|---|---|
| | **Males** | **Females** |
| Cortical Gray Matter (cGM) | 599 ± 57 | 563 ± 112 |
| White Matter (WM) | 358 ± 41 | 329 ± 101 |
| Cerebrospinal Fluid (CSF) | 219 ± 29 | 208 ± 43 |
| Extra-axial Cerebrospinal Fluid (ea-CSF) | 114 ± 15 | 114 ± 27 |
| Intracranial Volume (ICV) | 1471 ± 129 | 1373 ± 282 |
| Parasagittal Dura (PSD) | 5 ± 1.7 | 5 ± 2 |

### PSD and brain volumes

An example of PSD segmentation is represented in Fig. 3. The average PSD volume was $5 ± 2$ cm³. Significant correlations were identified between PSD volume and ea-CSF volume ($R = 0.33$; $p$-value = 0.0003), CSF volume ($R = 0.29$; $p$-value = 0.002) (Fig. 4). No correlations were identified between PSD volume and ICV, WM, or cGM (Table 3).

### PSD volume and clinical scores

PSD volume displayed an overall significant inverse relationship with IQ class ($p$-value = 0.0242, $F$-value = 3.071; one-way ANOVA) (Fig. 5), but not with ADOS-2 CSS scale ($p$-value = 0.126, $F$-value = 2.157; one-way ANOVA). Subsequent post-hoc analyses showed only a significant difference in the PSD volume between patients with normal and with severe IQ deficit scores ($p$-value = 0.022; one-tailed $t$-test). No other brain structure volume was correlated with clinical severity.

### PSD and ea-CSF volume correlation in developmental delay

In children with severe developmental delay, the PSD volume is smaller compared to those with normal IQ, despite having the same volume of ea-CSF. In other words, the correlation between PSD volume and ea-CSF fails to reach statistical significance in children with severe developmental delay ($R = 0.103$; $p$-value = 0.6), whereas, in children with normal IQ, this correlation appears to be statistically significant ($R = 0.515$; $p$-value = 0.02) (Fig. 6).

## Discussion

The role of PSD as a CSF-draining pathway is unexplored in both healthy developing children and ASD. We found a robust positive correlation between PSD volume, CSF, and ea-CSF volume and an inverse relationship between PSD volume and the severity of developmental delay, or IQ, in our cohort of ASD children. Severe developmental delay may be a consequence of an underdeveloped PSD which is inefficient in draining CSF, contributing thereby to the accumulation of toxic substances and promoting subtle neuroinflammatory processes often associated with ASD[25,41,42]. These

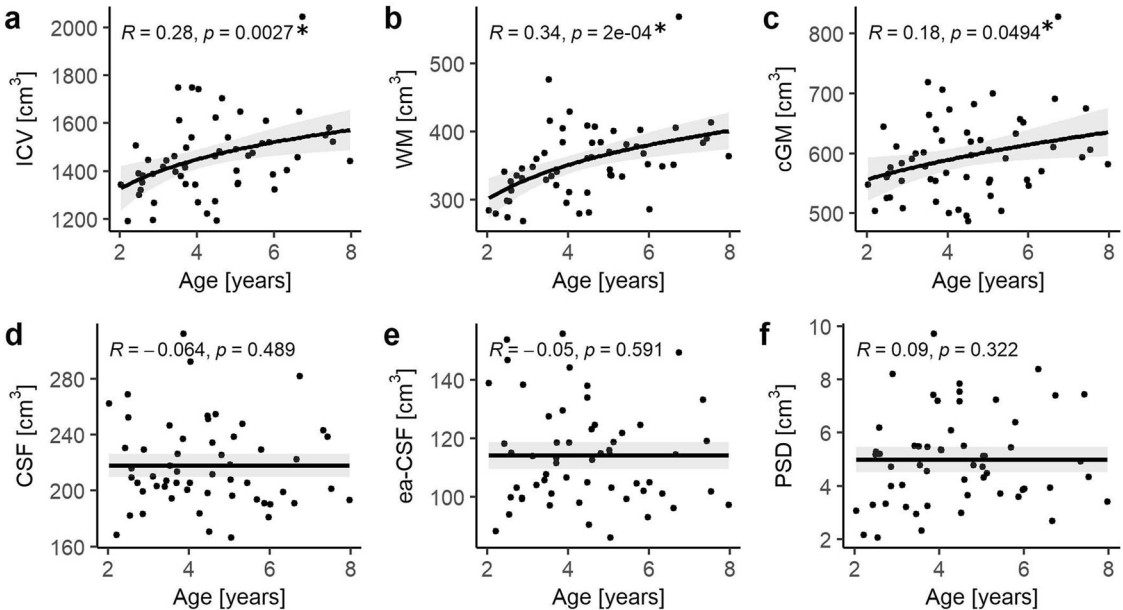

**Fig. 2 | Correlation of age and volume of different brain structures. a** intracranial volume (ICV), **b** white matter (WM), **c** cortical gray matter (cGM), **d** cerebrospinal fluid (CSF), **e** extra-axial cerebrospinal fluid (ea-CSF), and **f** parasagittal dura (PSD). The best model fits (logarithmic fit for ICV and WM; square fit for cGM and constant fit for all other brain structures) are reported as trend line (continuous line) with 95th confidence intervals. Volume/age relationships were quantified in terms of Kendall correlation coefficient ($R$) and $p$-value ($p$). Significant age-related correlations were observed with ICV and WM. After correcting for multiple comparisons no significant correlation was found between age and cGM. Number of patients ($N$) = 56.

**Fig. 3 | Example of segmented PSD.** The binary mask of the segmented PSD structure is highlighted in green in the three orthonormal planes and superimposed to the FLAIR sequence used in the segmentation. A 3D render of the cGM and the PSD structure is reported in the right panel.

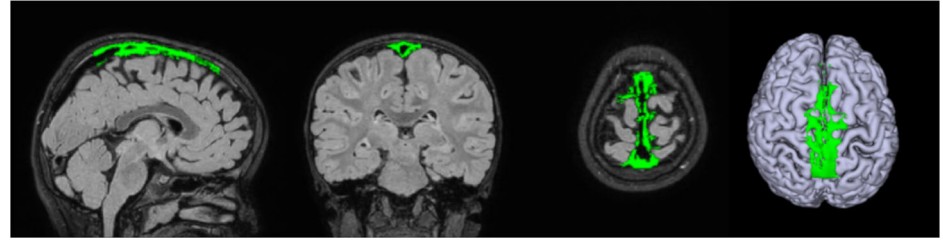

findings hold importance in light of the growing understanding of the role that PSD plays in promoting the drainage of CSF from the brain, in the removal of waste materials, and in facilitating immune-survellaince[1,25,43].

The ea-CSF has been described as the CSF space that envelops the cerebral dorsal subarachnoid space which contains CSF that is in direct proximity with the cerebral meninges and PSD[23]. It excludes the ventricular space and the lower or ventral portion of the subarachnoid space. Increased ea-CSF volume is a well-documented potential MRI biomarker in children with ASD and those at high risk of developing ASD[22,23,44]. Our findings add to the substantial body of literature that indicates altered CSF dynamics in this population[22,42].

In the traditional model, arachnoid granulations (AGs) are recognized as the primary sites of CSF absorption[45]. A recent study described five different types of AGs in the adult brain possessing different capacities for CSF transfer into MLs[46]. AGs typically reach maturity by the age of 18 months, but their numbers change over the lifespan[47,48]. In some individuals, AGs are completely absent without changes in CSF homeostasis, suggesting that there are alternative routes to CSF absorption, the PSD being one of them[47,49].

Age-related developmental trajectories for WM and cGM are well-documented in typically developing children with cGM showing an inverted U-shape growth trajectory compared to the WM that continues to increase till early adulthood[50,51]. In our cohort, WM, cGM, and ICV volumes

increased with age. This closely mirrors the developmental trajectories reported in a large longitudinal study on children with ASD and normally developing children[52]. The structural organization of the brain tissue and the maturation process of CSF production and absorption pathways in the developing brain are thought to affect CSF volume trajectories with age[53]. Recent works suggest that beyond the age of 4 years, no change in ea-CSF is observed in children with ASD[54,55]. Our work also confirms that the volume of ea-CSF does not change with age.

The development of meninges in the postnatal period reveals that MLs, including the meninges and the calvarium, continue to develop postnatally[56,57]. Although a few studies have explored PSD volume in healthy adults and individuals with neurodegenerative conditions, the PSD volume in both typically developing children and those with ASD has yet to be explored in the literature. Melin et al. [15] reported a PSD volume of $4.19 \pm 2.07 \, cm^3$ in a heterogeneous group comprising healthy adults and individuals with CSF disorders, whereas Song et al. reported an average PSD volume of $11.85 \pm 2.16 \, cm^3$ among adults diagnosed with Alzheimer's disease[15,17]. Therefore, although direct evidence is lacking, the growth trajectory of PSD in the developing brain is expected to follow the growth of the meninges, the dural venous system, and the calvarium in early childhood[58]. The volume of PSD did not correlate with the volumes of WM, cGM, or ICV but it strongly correlated with the volumes of CSF and ea-CSF. Although we report findings on children, they align with existing literature that has

**Fig. 4 | Correlation between PSD volume, extra-axial CSF (ea-CSF) and CSF. a** A significant positive correlation was observed between PSD volume and ea-CSF; **b** schematic representation of ea-CSF (blue) and segmented PSD (green); **c** significant positive correlation between PSD and CSF; **d** schematic representation of CSF (blue) and segmented PSD (green). Correlations were calculated using the Kendall correlation coefficient. Number of patients ($N$) = 56.

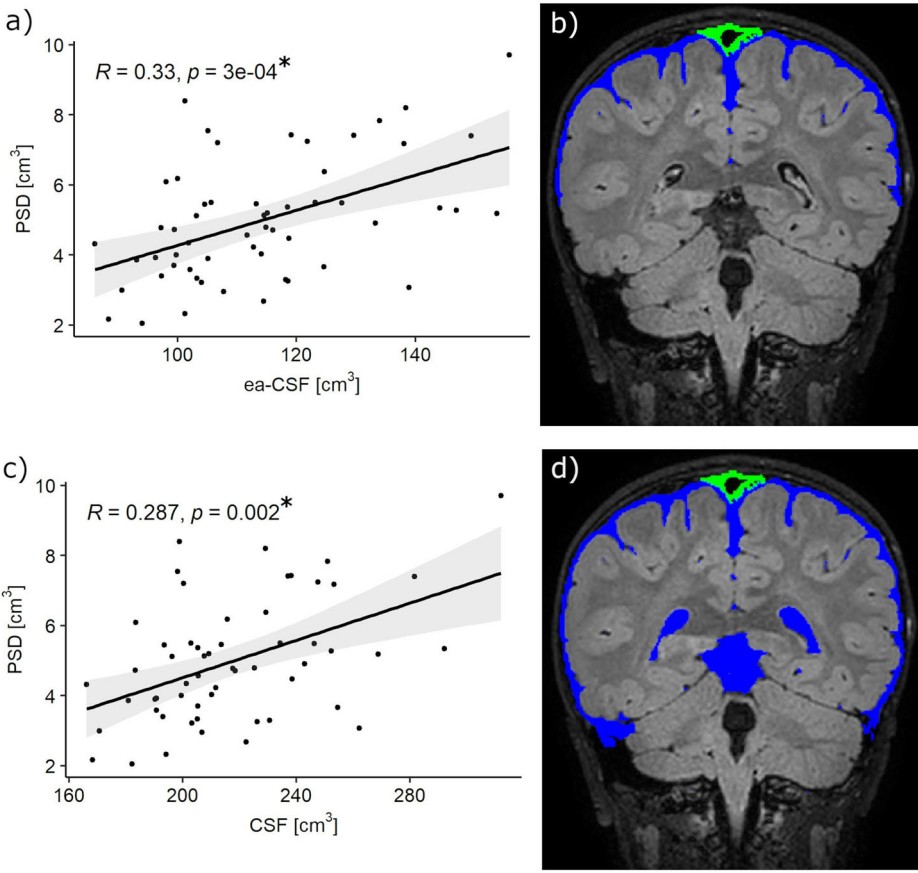

utilized similar deep learning-based algorithms to derive PSD volumes in adult humans over the age of 20 years[13,14,16]. These results further emphasize the crucial role of PSD in the exchange of CSF from the dorsal subarachnoid space.

In our study, PSD volume did not correlate with cGM or WM volumes[13,14]. Again, this finding is in line with literature on adults in which PSD volume was not correlated to age-related brain atrophy but rather only with CSF volume, underscoring the important link between CSF and PSD[16]. In a separate study involving patients with Alzheimer's disease, PSD volumes were significantly correlated with an increasing burden of amyloid beta deposition with no significant correlation observed with overall brain atrophy[17]. Furthermore, while studies in human adults reveal a significant positive association between PSD volume and age, in our study no age-related effect on PSD volume was observed notwithstanding changes in aforementioned brain volumes over age. Further studies are required to fully comprehend the normal development of PSD in the developing brain. In addition, there is little understanding of the relationship between PSD

### Table 3 | PSD volume and its correlation with brain morphological variables

| Brain structure | Correlation coefficient with PSD volume | P-value |
|---|---|---|
| Intracranial Volume (ICV) | 0.119 | 0.193 |
| White Matter (WM) | 0.126 | 0.170 |
| Cortical Gray Matter (cGM) | 0.009 | 0.921 |
| Cerebrospinal Fluid (CSF) | 0.287 | 0.002[a] |
| Extra-axial Cerebrospinal Fluid (ea-CSF) | 0.330 | 0.0003[a] |

[a]p-value < 0.01 (after Bonferroni correction for multiple comparisons).

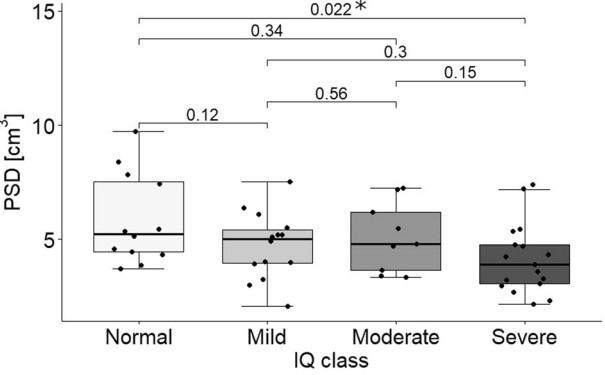

**Fig. 5 | PSD volume distribution in patients with different IQ classes (normal, mild, moderate, and severe).** The subsequent statistical comparisons (*t*-test) highlight a significant difference in PSD volume between patients with normal ($N$ = 12) and severe ($N$ = 17) IQ class (*p*-value = 0.022). Box limits indicate the range of the central 50% of the data, with a central line indicating the median value, and data points outside the upper and lower bounds are considered outliers.

volume and its CSF-draining capacity in very young children, and needs further investigation.

Another noteworthy finding in our study is the inverse relationship between PSD volumes and IQ scores among children with ASD. This observation implies that children with severe developmental delay also have a smaller PSD compared to children with normal IQ. The recent discoveries of the role of PSD and the MLs may shed some light on CSF dynamics that are altered in ASD[1,2,7,43]. The hypotrophic PSD in ASD children with severe developmental delay may harbor hypoplastic MLs, initiating a chain of

events that hampers CSF drainage, leads to the accumulation of cerebral toxins, and triggers neuroinflammatory processes affecting brain development[59]. Although an inverse relationship was found between PSD volume and the degree of developmental delay, it is noteworthy that CSF volume remains constant across various IQ levels (Fig. 7). No correlations were found with the ADOS-2 CSS scale.

It is well-known that PSD is not the only pathway for CSF efflux. Since the ea-CSF volume remained constant in children with normal and severe developmental delay, it is likely that CSF drains more effectively through other CSF-draining pathways. Previous investigations have underscored the primary involvement of PSD in neuroimmune functions, positing its role in CSF drainage as secondary. It is important to note that our study cohort predominantly consists of children with moderate to severe ASD, which limits our ability to establish meaningful correlations with milder forms of the condition. On the contrary, our discoveries unveil a Pandora's box, suggesting that the investigation of MLs in ASD could potentially unlock neuroinflammatory processes and alter CSF homeostasis in ASD.

Meningeal cells play crucial roles in guiding the development of ventricular radial glial cells, ensuring proper neuronal development, and are heavily involved in neuro-immune functions[60]. While the specific origin of PSD is unknown, it is likely that this tissue contains meningeal cells and meningeal stroma, as it lies within the two layers of the dura mater.

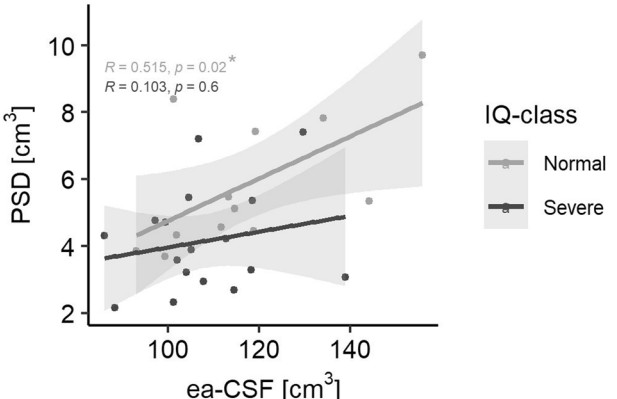

**Fig. 6 | Correlation between PSD volume and ea-CSF.** Children without developmental delay ($N = 12$) present a strong positive correlation between PSD and ea-CSF whereas in children with severe developmental delay ($N = 17$) this correlation does not reach statistical significance. Correlations were calculated using the Kendall correlation coefficient.

Meningeal neural progenitors migrate through multiple pathways within the brain parenchyma, contributing to cortical development, guiding neuronal connectivity, and forming membranes that delineate perivascular spaces around penetrating arterioles[61,62]. Findings from 16p11.2 mouse models of ASD clearly identify that endothelium is dysfunctional and affects the stability of blood vessels. This contributes to behavioral changes specific to ASD. Proper angiogenesis is also fundamental for optimal neurogenesis[63]. Anomalies in meningeal tissue development during the early stages of life, potentially influenced by genetic or epigenetic factors, may also contribute to established neuronal dysconnectivity in ASD. Our study suggests that PSD is underdeveloped in children with ASD who suffer more severe developmental delays. While additional investigations are necessary, it is proposed that a poorly developed PSD could potentially impact developmental processes, and promote neuroinflammation leading to dysregulation of neuronogenesis and angiogenesis.

In addition to PSD volumes, dilated perivascular spaces (DPVS) are considered indirect markers of obstructed drainage of fluids[64-67]. Only one study has examined DPVS in young children with ASD which reports a non-significant increase in the prevalence of DPVS in kids with severe form of ASD[68]. Our observations add to the literature whereby some form of obstruction to the movement of neurofluids may be present in ASD[10].

Our choice for employing 3D-FLAIR to segment PSD was based on previous initial work employing 3D-FLAIR for imaging MLs and quantifying the volume of PSD[1,3,15,69]. 3D-FLAIR is commonly used in the standard MRI protocol and is readily accessible. Neither contrast-enhanced T2-weighted black blood sequence nor sub-millimetric 3D-T2-weighted sequences that have been used in other studies to segment PSD were available in this retrospective study[13,14].

Our research paves the way for exploring newer avenues in the assessment of children with ASD, with the aim of identifying MRI markers suggestive of altered fluid dynamics and identifying treatment strategies. To achieve this, there are several critical steps to consider. Firstly, it is essential to establish quantitative measurements of PSD volumes in the developing brain. Secondly, the correlation between PSD volumes and serum-based markers of proinflammation should be investigated to gain deeper insights into the neuroinflammatory mechanisms involved. Thirdly, research into potential genetic alterations in children with ASD that could contribute to the underdevelopment of PSD and MLs warrants examination.

The primary constraint in our study is the lack of a reference group of typically developing children within our specified age range. There are no publicly available datasets in healthy young children that have 3D-FLAIR images that were employed for our deep learning algorithm for PSD segmentation, following established methods outlined in the work of Melin et al.[15]. Large datasets on children younger than 5 years is even more scarce.

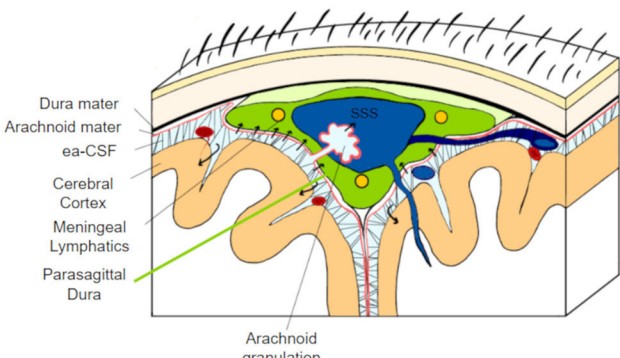

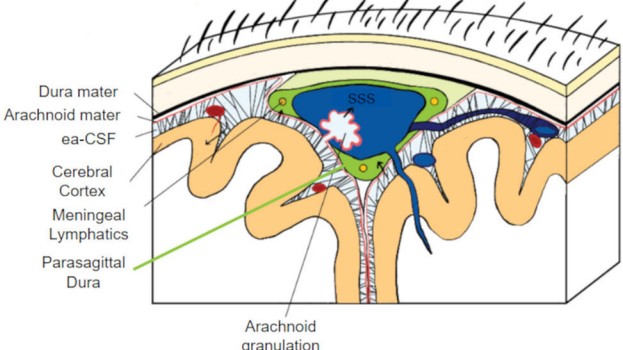

**Fig. 7 | Schematic representation of the parasagittal dura (PSD) in the coronal section in children with normal and severe IQ class.** In the left panel, a normal-appearing PSD is represented whereas in the right panel, an underdeveloped PSD is noted. Note that in both figures the volume of CSF remains constant. Black arrows represent the direction of movement of CSF. SSS: superior sagittal sinus.

Our institution's primary focus is on the diagnosis and treatment of very young children with moderate to severe neurodevelopmental disorders, which positions us favorably in acquiring MRI in young children with ASD. However, this specialized focus limits our ability to readily assemble a comparable group of healthy children. Nonetheless, this study provides greater insight and hope in our understanding of this devastating condition on the rise worldwide. Another limitation pertains to our sample size. We employed strict recruiting criteria to eliminate confounding factors. This still resulted in a sample size sufficient to detect significant associations between PSD volume, CSF, ea-CSF volume, and IQ class in ASD. A third limitation concerns the PSD segmentation pipeline. The scientific community lacks consensus on the methods, procedures, and even the types of images to be used for PSD segmentation. To address this, we utilized a cutting-edge segmentation method to generate the PSD mask from our images and incorporated a manual correction step to rectify any remaining errors. Furthermore, we confined the PSD segmentation to its central component to enhance the reproducibility of the process. Nevertheless, it is important to acknowledge that each of these choices may have an impact on the ultimate results. Lastly, we are aware that sedation can affect the dynamics of neurofluids. Animal research studies indicate that glymphatic clearance in the interstitial space increases during natural sleep and sedation with ketamine[70]. In another study, rats sedated with propofol exhibited an expansion of the extracellular space with improved interstitial fluid (ISF) drainage compared to other anesthetics like isoflurane[71]. While preclinical research suggests that sedation positively influences ISF drainage, human studies are lacking. It remains uncertain whether a 30-min sedation period in our study would result in significant changes in the PSD volume. This would be subject to further investigation. Even if we hypothesize an increased rate of CSF efflux into PSD during sedation, we would not expect our results to change because all our patients were under the same experimental condition during MR acquisition. Eide and Ringstad demonstrated that the rate of molecular clearance via PSD is not affected by sleep but there are no studies that determine changes in PSD volume with sleep or sedation in humans or in animals[72]. Successful implementation of MRI during deep sleep in young children would be ideal, however, this necessitates the creation of a specific ambiance and modification of the MRI protocol to allow for a shorter duration of the scan[73]. Such scanning poses practical challenges in our research studies, which remain currently difficult to navigate.

## Conclusions

This study suggests that an underdeveloped PSD may contribute to the severity of developmental delay in children with a diagnosis of ASD. Furthermore, PSD volume correlated only with total CSF and ea-CSF volume which validates its role in CSF drainage and strongly supports the emerging and ongoing revelation of CSF exchange between the subarachnoid space and the PSD.

## Data availability

All source data relative to images are available from the corresponding author on reasonable request. All source data to produce graphs in Figures 2, and 4–6 are available at https://doi.org/10.6084/m9.figshare.24582369.v4[74].

## Code availability

Codes and software versions used are available at https://doi.org/10.6084/m9.figshare.24582369.v4[74].

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

## Acknowledgements
This study was supported by the Italian Ministry of Health (Ricerca Corrente 2024) and 5 × 1000 funds for biomedical research.

## Author contributions
Project administration: N.A., L.L.; software - implementation of the computer code and supporting algorithms: L.L., G.R., T.C.; patient recruitment: F.L., E.M.; formal analysis and visualization: G.F., T.C., L.L., D.P., G.R.; Administration and writing of the original draft N.A.; writing – editing: G.F., L.L., T.C., D.P., G.R.; Supervision: R.C., M.M., D.P. All authors approved the final manuscript.

## Competing interests
The authors declare no competing interests.
