## [Peer Review File · Communications Medicine]

Reviewers' comments:

Reviewer #1 (Remarks to the Author):

The authors reliably quantified the volume of several anatomical structures in 48 patients with ASD and autism.

They found a correlation between PSD volume and developmental delay and Between PSD and extra-axial CSF.

Critics are summarized below to strengthen this manuscript.

The title seems vague and does not reflect the main findings of the manuscript. for example, the title could include something like the parasagittal dural volume correlates with low IQ and extra axial CSF in children with ASD and autism.

In the abstract It would be necessary to include the p values. as follows "but was significantly correlated with extra-axial CSF

volume (P = xx) and severity of developmental delay (P = XX)"

In the results section, it would be interesting to include control patients to compare the anatomical volumes of the different structures. If it is not possible a reasonable explanation should be included.

In Figure 2. It would be interesting to also include a representative example of extra-axial CSF segmentation (may be in red). This would help to better understand figure 3.

In table 3. The significance is established as * $p < 0.01$. It would be necessary to explain why particularly in the test it used $P < 0.01$ to indicate significance and not $P < 0.05$.

As a recommendation an anatomic representation of the PSD and the ea-csf would be useful to discuss the main findings and possible csf absorption problems associated with Autism.

Statistical analysis should include why Bonferroni correction used a $P < 0.01$ to be considered significant.

The conclusions seem vague and unspecific.

Reviewer #2 (Remarks to the Author):

The authors presented novel associations between parasagittal dura (PSD) volume, extra-axial CSF (EA-CSF) volume, and IQ in young male subjects ($n=48$) diagnosed with ASD. This is the first study to examine PSD volumes in young children and provides compelling further support for the growing theory that aberrant CSF dynamics play a role in neurodevelopmental disorders such as autism. The manuscript is clear, convincing, and timely given the increasing interest in CSF circulation and the glymphatic system. All that being said, there are some major and minor suggestions that I'd like to share with the authors.

Major comments:

1. The one citation offered for evaluating male and female subjects with ASD separately was insufficient justification for excluding females from this study. Since this is a retrospective study, it would be strongly encouraged to reanalyze the data with the inclusion of female subjects if available on hand. Even if it was not a sex-balanced design, it would greatly improve the impact of the findings. Otherwise, the title of the manuscript should be edited to reflect that this finding was only observed in males (i.e., "PSD is linked to CSF in young males with ASD").

2. Analysis: Is there any specific reason why the PSD – IQ relationship was analyzed using binned IQ classes rather than as a continuous variable? Did the findings still hold up when IQ was analyzed as continuous? Analyzing it as a continuous variable would increase power unless there is an underlying reason not to do so.

3. The authors found that underdeveloped PSD is linked with lower EA-CSF as well as severe developmental delay. The two associations combined seem relatively surprising and should be further expanded/interpreted in the Discussion.

a. For example, have the authors examined whether EA-CSF and IQ class relate? Given the findings of this study, should one expect greater EA-CSF in the normal IQ group rather than severe developmental delay? This seems like it might be at odds with the findings of Shen et al, though there are naturally significant differences between studies (and this current study didn't find a direct relationship between PSD and clinical severity).

b. Secondly, how would smaller PSD hamper CSF drainage (as noted in the Discussion), if smaller PSD volumes are also associated with lower EA-CSF volumes? There was also a similar trending but

not significant (after Bonferroni correction) association between PSD and general CSF volume. Further clarification on why EA-CSF would be lower in situations of possible impaired CSF drainage is needed.

Reviewer #3 (Remarks to the Author):

The link between neuronogenesis and vasculogenesis is tight during brain development. Numerous reviews (not cited in this paper) document both the neuroanatomic and possible molecular mechanisms between these two tightly linked processes. The premise of this paper is that anomalies in the PSD may indicate a process thought to be related to neuroinflammation, although it is unclear how generalized this pathophysiology is in all subgroups of autistic children. The intragroup comparisons in autism may yield some interesting findings such as a smaller PSD. However, without a age controlled TD group, it is difficult to know what to make of this data. Whether from local recruitment or databases, the TD information collected in an age dependent manner is critical to the interpretation of the data presented. More discussion about the normal development of the PSD during prenatal and post-natal periods is also needed for better context of the current findings. An anatomical diagram would also be helpful here as the average reader would not be familiar with the biology of the PSD. The authors also need to include a fuller discussion of vasculogenesis (known data) in autism. Human autopsy findings, findings in 16p11.2 mouse, and the writings of Vasudevan and Whitaker-Azmita are among the literature which should be reviewed in more detail.

We thank our reviewers for constructive feedback. Please find point-by-point responses to all comments here below.

Reviewer #1 (Remarks to the Author):

Q1: The title seems vague and does not reflect the main findings of the manuscript. for example, the title could include something like the parasagittal dural volume correlates with low IQ and extra axial CSF in children with ASD and autism.

Response: We change the title to better reflect the main findings of the manuscript “Parasagittal dural volume correlates with CSF volume and developmental delay in children with autism spectrum disorder”.

Q2: In the abstract It would be necessary to include the p values. as follows "but was significantly correlated with extra-axial CSF volume (P = xx) and severity of developmental delay (P = XX)"

Response: this phrase has been replaced with “but was significantly correlated with total CSF (p=0.002), extra-axial CSF volume (p=0.0003) and severity of developmental delay (p=0.024)” and added to the manuscript.

Q3: In the results section, it would be interesting to include control patients to compare the anatomical volumes of the different structures. If it is not possible a reasonable explanation should be included.

Response: We acknowledge that the lack of control subjects is a limitation of our study, and an explanation was given in the discussion section of the manuscript. We have further revised this paragraph which now reads as follows: The primary constraint in our study is the lack of a reference group of typically developing children within our specified age range. There are no publicly available datasets in healthy young children that have T2w-FLAIR images that was employed for our deep learning algorithm for PSD segmentation, following established methods outlined in the work of Melin et al (2023). Large datasets on children younger than 5 years is even more scarce. Requiring sedation to acquire MR images in healthy young children presents important ethical considerations and significant challenges.

Q4: In Figure 2. It would be interesting to also include a representative example of extra-axial CSF segmentation (may be in red). This would help to better understand figure 3.

Response: we thank the reviewer for this suggestion. We have added two figure parts to figure 3 that illustrate ea-CSF and CSF segmented areas.

Q5: In table 3. The significance is established as * $p < 0.01$. It would be necessary to explain why particularly in the test it used $P < 0.01$ to indicate significance and not $P < 0.05$.

Response: In the analysis comparing PSD volume with five distinct cerebral volumes (ICV, WM, GM, CSF, and ea-CSF), the Bonferroni correction was utilized to tackle the issue of multiple comparisons in statistical testing. This correction involved setting the p-value threshold at 0.01 (calculated as 0.05 divided by the number of comparisons), ensuring a more stringent criterion for determining statistical significance in each individual comparison. This explanation has now been added to the methods section and better reported in table 3 “: p-value < 0.01 (after Bonferroni correction for multiple comparisons)”*

Q6: As a recommendation an anatomic representation of the PSD and the ea-CSF would be useful to discuss the main findings and possible csf absorption problems associated with Autism.

Response: the below manual drawing of the anatomical structures has been created and added to the discussion section.

Fig. 6: Schematic representation of the parasagittal dura mater (PSD) in the coronal section in children with normal and severe IQ. In the former, a normal appearing PSD is represented whereas in the latter an underdeveloped PSD is noted. Note that in both figures the volume of CSF remains constant. Black arrows represent the direction of movement of CSF.

Q7: Statistical analysis should include why Bonferroni correction used a $P < 0.01$ to be considered significant.

Response: as above, in the analysis comparing PSD volume with five distinct cerebral volumes (ICV, WM, GM, CSF, and ea-CSF), the Bonferroni correction was utilized to tackle the issue of multiple comparisons in statistical testing. This correction involved setting the p-value threshold at 0.01 (calculated as 0.05 divided by the number of comparisons), ensuring a more stringent criterion for determining statistical significance in each individual comparison. This information is now reported in the main text of the manuscript.

Q8: The conclusions seem vague and unspecific.

Response: we have rewritten our conclusions to better summarize the results of this paper. The new conclusions are: This study suggests that an underdeveloped PSD may contribute to the severity of developmental delay in children with a diagnosis of ASD. Furthermore, PSD volume correlated only with CSF and ea-CSF volume which validates its role in CSF drainage and strongly supports the emerging and ongoing revelation of CSF exchange between the subarachnoid space and the PSD.

Reviewer #2 (Remarks to the Author):

The authors presented novel associations between parasagittal dura (PSD) volume, extra-axial CSF (EA-CSF) volume, and IQ in young male subjects (n=48) diagnosed with ASD. This is the first study to examine PSD volumes in young children and provides compelling further support for the growing theory that aberrant CSF dynamics play a role in neurodevelopmental disorders such as autism. The manuscript is clear, convincing, and timely given the increasing interest in CSF circulation and the glymphatic system. All that being said, there are some major and minor suggestions that I'd like to share with the authors.

Response: we thank the reviewer for appreciating our work.

Major comments:

Q1. The one citation offered for evaluating male and female subjects with ASD separately was insufficient justification for excluding females from this study. Since this is a retrospective study, it would be strongly encouraged to reanalyze the data with the inclusion of female subjects if available on hand. Even if it was not a sex-balanced design, it would greatly improve the impact of the findings. Otherwise, the title of the manuscript should be edited to reflect that this finding was only observed in males (i.e., “PSD is linked to CSF in young males with ASD”).

Response: we thank the reviewer for this important suggestion. We have now included the 8 female subjects to our cohort in our statistical analysis (total number of patients is now 56). A new table 1 reflects this change. With this modification, our results now appear even more robust. Multivariate analysis was not required since the best linear model for the PSD volumes distribution included only the ea-CSF volume as independent variable. No additional information would have been gained. All tables and figures have been revised. The modifications are as follows:

- **Correlation of age and volume of different brain structures (new Table 2 and Fig. 1):**
 1. Total WM and ICV volume continue to correlate positively with age and shows an even higher statistical significance (WM: $p=0.0002$ with respect to $p=0.004$; ICV: $p = 0.003$ with respect to 0.03).
 2. CSF, ea-CSF and PSD volumes continue to remain unchanged with age.
- **PSD volume and its correlation with brain morphological variables (new figure 3 and Table 3 have been added):**
 1. A greater statistical significance is now observed between PSD and ea-CSF ($p = 0.0003$ with respect to 0.005).
 2. With respect to past results, PSD volume now appears to significantly correlate with total CSF volume ($p = 0.002$).
- **PSD volume and clinical scores (new figure 4 has been added):** PSD volume continue to significantly correlate negatively with developmental delay ($p = 0.0242$).

Q2. Analysis: Is there any specific reason why the PSD – IQ relationship was analyzed using binned IQ classes rather than as a continuous variable? Did the findings still hold up when IQ was analyzed as continuous? Analyzing it as a continuous variable would increase power unless there is an underlying reason not to do so.

Response: In clinical setting, DSM-V provides guidelines for classifying developmental delay into four main groups: mild, moderate, severe, and extreme. Since this is a retrospective study, the division into four main groups of functioning (normal or delayed with its subtypes) also allowed us to include those patients for whom the psychometric limitation of the scales administered did not allow us to quantify the outcome in numerical terms.

Q3. The authors found that underdeveloped PSD is linked with lower EA-CSF as well as severe developmental delay. The two associations combined seem relatively surprising and should be further expanded/interpreted in the Discussion.

Response: Our updated findings, now encompassing female subjects, reveal an even more robust correlation between PSD volume and ea-CSF volume. In addition, PSD now strongly correlates with total CSF as well. This statistically significant association persists only with CSF related brain morphological measure, underscoring the evidence that the PSD plays a substantive role in CSF drainage. Although a negative correlation was found between PSD volume and the degree of developmental delay, it is noteworthy that ea-CSF volume remains constant across various IQ levels (see figure below. If needed, we can add it to the supplementary material). In other words, for the same volume of ea-CSF, PSD volume is smaller in children with severe IQ with respect to children with normal IQ. To illustrate this, we have added a new figure 5. It is very likely that alternative CSF drainage pathways function to compensate for an underdeveloped PSD which may form the basis of similar CSF volume across all classes of IQ. These results add to the growing evidence that PSD may not be a primary route of CSF efflux but rather act as a neuroimmune hub, as elucidated by Melin et al. in 2020 and Rustenhoven et al. 2022. Both references have been reported in the manuscript.

Supplementary figure: ea-CSF volume distribution in patients with different IQ classes (normal, mild, moderate and severe). The subsequent statistical comparisons (t-test) show no significant differences in ea-CSF volume among different IQ classes. P-values are noted in the graph.

Fig. 5: Correlation between PSD volume and ea-CSF: children without developmental delay present a strong positive correlation between PSD and ea-CSF whereas in children with severe developmental delay this correlation does not reach statistical significance.

Q4. Have the authors examined whether EA-CSF and IQ class relate?

Response: Yes. No relationship was observed between ea-CSF and the four different IQ classes. See response to Q3 (see supplementary figure above).

Q5. Given the findings of this study, should one expect greater EA-CSF in the normal IQ group rather than severe developmental delay? This seems like it might be at odds with the findings of Shen et al, though there are naturally significant differences between studies (and this current study didn't find a direct relationship between PSD and clinical severity).

Response: Research conducted by Shen indicates increased ea-CSF volume in very young children at a high risk of developing autism, as well as in those with severe autism. In contrast, our study reveals no correlation between ea-CSF and the degree of developmental delay (supplementary figure). In other words, we would not necessarily expect larger ea-CSF volume in patients with severe developmental delay. The negative correlation between PSD and developmental delay however suggests, that children with severe developmental delay have a smaller PSD (new fig. 5).

Q6. Secondly, how would smaller PSD hamper CSF drainage (as noted in the Discussion), if smaller PSD volumes are also associated with lower EA-CSF volumes? There was also a similar trending but not significant (after Bonferroni correction) association between PSD and general CSF volume. Further clarification on why EA-CSF would be lower in situations of possible impaired CSF drainage is needed.

Response: In addition to ea-CSF volume, our updated findings unequivocally establish a correlation between PSD volume and total CSF volume ($p=0.002$). There is little understanding of whether a larger PSD should also necessarily be a normo-functioning tissue even though previous studies have suggested a possible compensatory role of a hypertrophied PSD in aging adults. We report the significant correlation of PSD with the only brain morphological parameter, i.e. CSF, of direct relevance. It is well-known that PSD is not the only pathway for CSF efflux and our results suggests the likelihood of effective drainage through other CSF-draining pathways. Previous investigations have underscored the primary involvement of the PSD in neuroimmune functions, positing its role in CSF drainage as secondary. While further investigations are needed, it is suggested that PSD may affect processes of developmental delay from a neuroinflammatory point of view and not represent causative role in the development of autism spectrum disorder. The complex dynamics of neurofluids and lack of normative data on young children, makes it difficult to draw definitive conclusions. Further investigations may be necessary to clarify this concept.

Reviewer #3 (Remarks to the Author):

Q1: The link between neuronogenesis and vasculogenesis is tight during brain development. Numerous reviews (not cited in this paper) document both the neuroanatomic and possible molecular mechanisms between these two tightly linked processes. The premise of this paper is that anomalies in the PSD may indicate a process thought to be related to neuroinflammation, although it is unclear how generalized this pathophysiology is in all subgroups of autistic

children. The intragroup comparisons in autism may yield some interesting findings such as a smaller PSD.

Response: Our study was designed to identify potential alterations in the parasagittal dura among children with Autism Spectrum Disorder (ASD). The parasagittal dura (PSD), recently recognized as a tissue facilitating cerebrospinal fluid (CSF) drainage from the subarachnoid space, holds significance as a key neuroimmune hub. Recent studies have shown that PSD volume increases with the aging process in humans and exhibits a positive correlation with an elevated load of amyloid in Alzheimer's disease patients. Preliminary data on adult human beings suggest that PSD may play a crucial role in immune cell trafficking and contribute to neuroinflammatory processes. We therefore focused our attention to PSD tissue. Our study cannot establish connections between neuronogenesis and vasculogenesis, as such aspects are beyond the intended scope of our investigation. It is essential to note that there are no studies that shed light on the role of PSD in children, adding to the distinctive nature of our exploration.

Q2: However, without a age controlled TD group, it is difficult to know what to make of this data. Whether from local recruitment or databases, the TD information collected in an age dependent manner is critical to the interpretation of the data presented.

Response: we recognize that the absence of data from normal healthy subjects is a limitation. As mentioned in the limitation section of our study, there is no data that on healthy subjects specially under the age of 5. Above the age of 5, large downloadable datasets do not contain FLAIR images that were used to segment PSD.

Q3. More discussion about the normal development of the PSD during prenatal and post-natal periods is also needed for better context of the current findings.

Response: This would be very appropriate but unfortunately there is no data on PSD development in human beings or in animal models. Some information on meningeal development is detailed in the discussion session.

Q4. An anatomical diagram would also be helpful here as the average reader would not be familiar with the biology of the PSD.

Response: we have created one and added to the text

Q5. The authors also need to include a fuller discussion of vasculogenesis (known data) in autism. Human autopsy findings, findings in 16p11.2 mouse, and the writings of Vasudevan and Whitaker-Azmita are among the literature which should be reviewed in more detail.

Response: we believe that this is beyond the scope of our work. We have based our work on investigating the parasagittal dura and not on vessels.

Reviewers' comments:

Reviewer #1 (Remarks to the Author):

The manuscript has been significantly improved, and all concerns have been addressed. It is now suitable for publication.

Reviewer #2 (Remarks to the Author):

I greatly appreciate the authors for being very receptive to feedback and suggestions. It was very encouraging to see that the results did not change, and in fact became slightly stronger, with the addition of 8 female subjects. This seems to be indicative of the stability of the results and that this finding is likely not sex-specific. Some final, minor remarks:

Minor comments:

1. In Fig. 6, the schematic presents “normal vs severe IQ”, and I believe this wording should be clarified. The “severe IQ” label is very unclear when not in context, and it should be made clearer that this is referring to developmental delay classes.
2. Typos throughout, for example: “all subjects underwent brain MRI examinations, as well as etiologicinstrumental” and “2 or greater than 8 yearsand”
3. More detail is needed about the scanning conditions in the study; were all the subjects sedated? If not, was this collected under natural sleep conditions or were the subjects awake?
4. The authors added in a statement “Requiring sedation to acquire MR images in healthy young children presents important ethical considerations and significant challenges” as one of the reasons that there is no normative comparison group, but there are countless examples of successful natural sleep, non-sedated imaging studies in children under 5. If the children in this study are sedated (it was unclear to me whether that was the case, see point 3 above), do the authors believe that sedated versus non-sedated scans would make an impact on studying the PSD? Please expand on this.

Reviewer #3 (Remarks to the Author):

The authors have really not answered my questions in my original review. My suggestions remain the same.

Reviewer #4 (Remarks to the Author):

Title: Parasagittal dural volume correlates with CSF volume and developmental delay in children with autism spectrum disorder

Summary:

The article examines the correlation between parasagittal dura (PSD) volume, brain structure volumes and clinical severity scales in young children with autism spectrum disorder (ASD). The authors employed a Deep Learning based approach to extract PSD volume from 3D-T2 Fluid Attenuated Inversion Recovery sequences. The data was then analyzed using statistical analysis and Kendall correlation tests were used for determining correlations.

Overall comments:

This review focuses on the deep learning-based model used in the revised article for PSD volume extraction. The authors do not provide sufficient information including performance metrics from the employment of the deep learning-based model or heterogeneity of the data needed for reliability. The authors should describe in detail the performance of the model on test data. The authors mentioned that resulting segmentations were manually refined by the neuroradiologist, but they should clarify how many erroneous segmentations were there. This is particularly important because the training sample was very small as compared to the other study referenced here. The authors address their limited sample size in the discussion section, but there needs to be further discussion on data augmentation and synthetic data usage for addressing such limitation and the justification for not using any such methods. There is no reference #69 as indicated in the article (line 354). The authors should review their references to ensure appropriate citation.

Abstract:

Lines 11-12: Authors should mention here which “Deep Learning-based approach” they used to extract PSD volume from 3D-T2 Fluid Attenuated Inversion Recovery sequences.

Results:

Lines 103-107: An example of PSD segmentation is provided in figure 2 with quantitative information on average PSD volume. The authors should provide background information on the expected PSD volume.

More importantly, the performance metrics from testing should be provided here from the employment of U-Net for extracting PSD volume.

Methods:

Lines 346-348: The authors mentioned in Reference #67 that “convolutional neural networks (CNNs), particularly the U-Net architecture and its variants, have become the state-of-the-art approach to perform an automatic and user independent segmentation.”

The cited article is only focused on fetal MR image segmentation while indicating the dominance of U-Net backbone. It also emphasizes the need for heterogenous data sets for reliable results. The authors should include more convincing arguments for selecting U-net for their particular segmentation and elaborate on the heterogeneity of their data.

Their reference #66 discusses “U-Net and Its Variants for Medical Image Segmentation” and describes U-Net “as the “primary tool for segmentation tasks in medical imaging”, which is not cited here, but will be more appropriate here. The authors should recheck their citations for any discrepancies.

Lines 348-351: The authors mentioned the development of an in house 2D U-Net backbone-based architecture. The article cited here used at least 20-35 images for training their U-Net. How did the authors determine that their small training set including 10 healthy subjects was reliable? The authors need to provide the performance metrics following testing on the 48 ASD patients to assess whether the model was appropriate.

Lines 351-352: “All the resulting segmentations were manually refined by the neuroradiologist to correct for erroneous segmentations.”

The authors should describe what was considered as “erroneous segmentations” and how many segmentations contained such errors to assess the model performance.

Lines 352-354: “The U-Net backbone based architecture was employed given the constrained training dataset, as implemented in previous PSD segmentation studies 69. “

There is no reference #69 provided in this paper. The authors should recheck their references for accuracy.

We thank our reviewers for constructive feedback. Please find point-by-point responses to all comments here below.

Reviewer #1 (Remarks to the Author):

The manuscript has been significantly improved, and all concerns have been addressed. It is now suitable for publication.

Response: Thank you.

Reviewer #2 (Remarks to the Author):

I greatly appreciate the authors for being very receptive to feedback and suggestions. It was very encouraging to see that the results did not change, and in fact became slightly stronger, with the addition of 8 female subjects. This seems to be indicative of the stability of the results and that this finding is likely not sex-specific. Some final, minor remarks:

Response: We thank the reviewer for this excellent suggestion.

Minor comments:

Q1: In Fig. 6, the schematic presents “normal vs severe IQ”, and I believe this wording should be clarified. The “severe IQ” label is very unclear when not in context, and it should be made clearer that this is referring to developmental delay classes.

Response: We have labeled Fig. 6 as follows: Normal IQ class vs. Severe IQ class. We have also added the label PSD which was missing in the previous figure (see figure reported here below). In addition, we have added the scores that define the four IQ classes in the methods section.

Q2: Typos throughout, for example: “all subjects underwent brain MRI examinations, as well as etiologicinstrumental” and “2 or greater than 8 yearsand”

Response: This is unfortunate and we apologize for the typos. We have made the necessary corrections and went through the document multiple times to check for typos.

Q3: More detail is needed about the scanning conditions in the study; were all the subjects sedated? If not, was this collected under natural sleep conditions or were the subjects awake?

Response: All our patients were under sedation throughout the entire MR examination. The following statement has been added to the methods section:

“All our participants were sedated with continuous intravenous infusion of propofol.”

Q4: The authors added in a statement “Requiring sedation to acquire MR images in healthy young children presents important ethical considerations and significant challenges” as one of the reasons that there is no normative comparison group, but there are countless examples of successful natural sleep, non-sedated imaging studies in children under 5.

Response: We agree with the reviewer's assessment that young, healthy children can undergo MRI scans while naturally asleep. For clinical diagnostic purposes, newborns are typically scanned in spontaneous sleep condition. To achieve this condition, newborn babies' feeding is timed appropriately to induce deep sleep, followed by immediate scanning to enhance the likelihood of a

successful MRI. In fact, in our other ethically approved research studies, we regularly perform MRI scans on newborns under two months of age during natural sleep. However, in our experience, obtaining high-quality diagnostic images in babies above three months of age, often requires sedation to ensure a minimum of 25-30 minutes scan duration with diagnostic quality. Maintaining natural sleep throughout the scan involves specific modifications to MRI sequence parameters aimed at reducing ambient noise (such as reducing slew rate and gradient amplitudes) and shortening acquisition times, albeit at the expense of reduced spatial resolution. The successful implementation of MRI during deep sleep necessitates the creation of a specific ambiance and modifying the MRI protocol, as proposed by Dean et al. in Pediatric Radiology 2014. Nonetheless, such scanning poses ethical and practical challenges in our research studies, which remain currently difficult to navigate.

We have added the following sentences in the discussion section.

“Successful implementation of MRI during deep sleep in young children would be ideal, however this necessitates the creation of a specific ambiance and modification of the MRI protocol to allow for a shorter duration of the scan (Dean 2014). Such scanning poses ethical and practical challenges in our research studies, which remain currently difficult to navigate.”

Q5: If the children in this study are sedated (it was unclear to me whether that was the case, see point 3 above), do the authors believe that sedated versus non-sedated scans would make an impact on studying the PSD? Please expand on this.

Response: All our participants were sedated with continuous intravenous infusion of propofol. Animal research studies indicate that glymphatic clearance in the interstitial space increases during natural sleep and sedation with ketamine (Benveniste et al., 2019). In another study, rats sedated with propofol exhibited an expansion of the extracellular space with improved interstitial fluid (ISF) drainage compared to other anesthetics like isoflurane (Zhao et al., 2021). While preclinical research suggests that sedation positively influences ISF drainage, human studies are lacking. Eide and Ringstad demonstrated that the rate of molecular clearance via PSD is not affected by sleep but there

are no studies that determine changes in PSD volume with sleep or sedation in humans or in animals (Eide et al. 2021). It remains uncertain whether a 30-minute sedation period would result in significant changes in the PSD volume. This would be subject to further investigation. Even if we hypothesize an increased rate of CSF efflux into PSD during sedation, we would not expect our results to change because all our patients were under the same experimental condition during MR acquisition.

We have added the above information in the limitations section.

Reviewer #3 (Remarks to the Author):

The authors have really not answered my questions in my original review. My suggestions remain the same.

Response: We apologize for the unsatisfactory response to your suggestions and questions. We have added some new comments here below.

Q1: The link between neuronogenesis and vasculogenesis is tight during brain development. Numerous reviews (not cited in this paper) document both the neuroanatomic and possible molecular mechanisms between these two tightly linked processes. The premise of this paper is that anomalies in the PSD may indicate a process thought to be related to neuroinflammation, although it is unclear how generalized this pathophysiology is in all subgroups of autistic children. The intragroup comparisons in autism may yield some interesting findings such as a smaller PSD.

Response: Our study was designed to identify potential alterations in the parasagittal dura (PSD) among children with Autism Spectrum Disorder (ASD). PSD is recognized as a tissue facilitating CSF drainage from the subarachnoid space, and has been proposed as a key neuroimmune hub in

human adults (Melin et al., 2023). But, the role of PSD in neurogenesis and in vasculogenesis in the prenatal and postnatal phases of human life has not been studied thus far. However, as suggested by the reviewer, previous works of Vasudevan and Whitaker-Azmita have demonstrated that meningeal progenitor cells play a crucial role in embryonal and fetal neurogenesis and vasculogenesis. Since, PSD is a tissue that lies within the dural meningeal layer, it could be speculated that a maldeveloped PSD could affect neuronal and vascular development. We agree that subgroup analysis (such as syndromic vs idiopathic ASD) may provide useful information in the future.

We have added the following paragraph in the discussion section of our manuscript.

“Meningeal cells play crucial roles in guiding the development of ventricular radial glial cells, ensuring proper neuronal development, and are heavily involved in neuro-immune functions⁴⁶. While the specific origin of PSD is unknown, it is likely that this tissue contains meningeal cells and meningeal stroma, as it lies within the two layers of the dura mater. Meningeal neural progenitors migrate through multiple pathways within the brain parenchyma, contributing to cortical development, guiding neuronal connectivity, and forming membranes that delineate perivascular spaces around penetrating arterioles^{47,48}. Findings from 16p11.2 mouse models of ASD clearly identify that endothelium is dysfunctional and affects stability of blood vessels. This contributes to behavioural changes specific to ASD. Proper angiogenesis is also fundamental for optimal neurogenesis⁴⁹. Anomalies in meningeal tissue development during early stages of life, potentially influenced by genetic or epigenetic factors, may also contribute to established neuronal dysconnectivity in ASD. Our study suggests that PSD is underdeveloped in children with ASD who suffer more severe developmental delay. While additional investigations are necessary, it is proposed that a poorly developed PSD could potentially impact developmental processes, promote neuroinflammation leading to dysregulation of neurogenesis and angiogenesis.”

Q2: However, without a age controlled TD group, it is difficult to know what to make of this data. Whether from local recruitment or databases, the TD information collected in an age dependent manner is critical to the interpretation of the data presented.

Response: We recognize that the absence of data from normal healthy subjects is a limitation. As mentioned in the limitation section of our study, there is no FLAIR data on healthy subjects specially under the age of 5 including large downloadable datasets. This limitation has been detailed in the limitation section of the manuscript.

Q3: More discussion about the normal development of the PSD during prenatal and post-natal periods is also needed for better context of the current findings.

Response: PSD is a newly discovered tissue without much data on its precise embryonal and fetal development. We find it hard to make comparisons between PSD growth and meningeal development. Please also see our response to Q1 above.

Q4: An anatomical diagram would also be helpful here as the average reader would not be familiar with the biology of the PSD.

Response: We have created one and added to the text. Please see Fig. 6.

Q5: The authors also need to include a fuller discussion of vasculogenesis (known data) in autism. Human autopsy findings, findings in 16p11.2 mouse, and the writings of Vasudevan and Whitaker-Azmita are among the literature which should be reviewed in more detail.

Response: We have read works by authors suggested by the reviewer. The following statement has been incorporated in the discussion section, together with the response to Q1 above.

“Findings from 16p11.2 mouse models of ASD clearly identify that endothelium is dysfunctional and affects stability of blood vessels. This contributes to behavioral changes specific to ASD. Proper angiogenesis is also fundamental for optimal neurogenesis.”

Reviewer #4 (Remarks to the Author):

This review focuses on the deep learning-based model used in the revised article for PSD volume extraction.

Q1: Lines 11-12: Authors should mention here which “Deep Learning-based approach” they used to extract PSD volume from 3D-T2 Fluid Attenuated Inversion Recovery sequences.

Response: We have added this information in the abstract of our manuscript.

“We employed a semi-supervised two step pipeline to extract PSD volume from 3D-T2 Fluid Attenuated Inversion Recovery sequences, based on U-Net followed by manual refinement of the extracted PSD masks.”

Q2: Lines 103-107: An example of PSD segmentation is provided in figure 2 with quantitative information on average PSD volume. The authors should provide background information on the expected PSD volume. More importantly, the performance metrics from testing should be provided here from the employment of U-Net for extracting PSD volume.

Response: In the Discussion section we have incorporated background information concerning existing literature regarding PSD volume. To the best of our knowledge, there are currently no studies exploring the PSD in both healthy and ASD children. The following statement has been added in the discussion section.

“Although a few studies have explored PSD volume in healthy adults and individuals with neurodegenerative conditions, the PSD volume in both typically developing children and those with ASD has yet to be explored in the literature. Melin et al. (2023) reported a PSD volume of $4.19 \pm 2.07 \text{ cm}^3$ in a heterogenous group comprising healthy adults and individuals with CSF disorders, whereas Song et al. reported an average PSD volume of $11.85 \pm 2.16 \text{ cm}^3$ among adults diagnosed with Alzheimer's disease.”

Concerning the performance metrics for PSD segmentation, please see response to Q6 below.

Q3: Lines 346-348: The authors mentioned in Reference #67 that “convolutional neural networks (CNNs), particularly the U-Net architecture and its variants, have become the state-of-the-art approach to perform an automatic and user independent segmentation.”

The cited article is only focused on fetal MR image segmentation while indicating the dominance of U-Net backbone. It also emphasizes the need for heterogenous data sets for reliable results. The authors should include more convincing arguments for selecting U-net for their particular segmentation and elaborate on the heterogeneity of their data.

Response: We added better references providing a generic review of the U-NET applications to the medical image segmentation framework.

The U-Net architecture is commonly utilized for segmentation tasks, including the segmentation of anatomical structures like the parasagittal dura, thanks to its ability to identify highly detailed structure contours and to disentangle touching objects (Siddique et al., 2021). Furthermore, U-Net performs well even with limited training data (Guo et al., 2022), which is often the case in medical imaging due to the challenges associated with acquiring labeled datasets, thanks to the possibility to include data augmentation in the training procedure. Nonetheless, it has already been used to segment the PSD volume by Ringstad and colleagues in different papers (Ringstad et al., 2021; Melin et al., 2023).

We agree with the referee that heterogeneity in the data is fundamental when developing a generic tool able to segment the structure of interest over different images, different contrasts and different sequence set ups. Unluckily, we do not have such kind of dataset. On the contrary, in this work we used the U-Net network to support and speed up the segmentation of the PSD on a specific problem, i.e. on a population of children with ASD, using 3D-FLAIR images acquired with our scanner with a specific sequence. The trained network won't generalize over different kind of images, but it works

very well on our problem (see the new images in the supplementary material). We are aware that there is a risk of overfitting of the data, therefore we added a manual correction of the extracted PSD mask to limit this potential issue.

Q4: Their reference #66 discusses “U-Net and Its Variants for Medical Image Segmentation” and describes U-Net “as the “primary tool for segmentation tasks in medical imaging”, which is not cited here, but will be more appropriate here. The authors should recheck their citations for any discrepancies.

Response: We apologize for discrepancies present in the manuscript. We have revised our citations and made the necessary corrections in the manuscript. The correct reference for U-Net and Its Variants for Medical Image Segmentation is as follows:

“In this context, convolutional neural networks(CNNs), particularly the U-Net architecture and its variants, have become the state-of-the-art approach to perform an automatic and user independent segmentation^{71,72}. Thus, we developed an in house 2D U-Net backbone-based architecture with an intent to facilitate the segmentation process⁷³.”

Q5: Lines 348-351: The authors mentioned the development of an in house 2D U-Net backbone-based architecture. The article cited here used at least 20-35 images for training their U-Net. How did the authors determine that their small training set including 10 healthy subjects was reliable?

Response: U-Net was trained on a minimum of 150 images per healthy subject resulting in a set of 2250 training images in the coronal section.

We have added this sentence in the Materials and Methods section of our manuscript:

“Each subject dataset comprised of at least 150 images for a total training set of 2250 coronal images. This U-Net was then applied on our cohort of 56 children with ASD.”

Q6: The authors need to provide the performance metrics following testing on the 48 ASD patients to assess whether the model was appropriate.

Response: The U-Net was trained on healthy adults and applied on 56 ASD children (48 males plus 8 new female subjects). We apologize for not clarifying this in our revised manuscript. We originally tested the U-Net on 10 ASD children (outside of the current cohort). The performance metrics between manual segmentation and automatic segmentation were assessed in terms of DICE-score and are reported in the figure below. With a DICE-score of 0.73 ± 0.19 , the model was deemed suitable and thus employed as a tool to facilitate the segmentation process within our cohort.

The above graph illustrates DICE-score histogram and density distribution. DICE-score was calculated considering manual segmentation as ground truth and automatic segmentation of the test set.

In addition, we would like to clarify that our primary objective did not contemplate the introduction of a novel and validated method to automatically segment the PSD. Rather, our focus was to utilize deep learning as an auxiliary tool to facilitate the segmentation process. The resulting masks were

revised and manually corrected by a neuroradiologist to ensure accuracy (as performed by Melin et al., 2023).

Q7: Lines 351-352: “All the resulting segmentations were manually refined by the neuroradiologist to correct for erroneous segmentations.”

The authors should describe what was considered as “erroneous segmentations” and how many segmentations contained such errors to assess the model performance.

Response: A figure (Fig. S1) has been added to the manuscript in the Supplementary Materials section, illustrating examples of the disparities between the automated segmentation generated by the U-Net and the final mask, which was further refined manually. These refined masks were employed for the computation of PSD volume and subsequent analysis.

Moreover, Fig. S2 has been included in the Supplementary Materials section, depicting the histogram and density distribution of DICE-scores comparing the manually corrected segmentation of PSD as the ground truth with automatic segmentation of PSD across the 56 children with ASD included in the analysis.

■ Automatic segmentation
 ■ Manual segmentation
 ■ Automatic and manual segmentation

Fig. S1: The above images illustrate automatic segmentation masks of PSD overlapped with manually corrected masks from ten randomly selected images from the entire cohort. Each example shows the cropped FLAIR image on the left side, while on the right side, FLAIR image is displayed overlapped

with both manual and automatic segmentations. Additionally, the DICE-score corresponding to each example is presented.

Fig. S2: The graph illustrates the histogram and density distribution of DICE-scores, comparing the manually corrected segmentation of PSD as the ground truth with automatic segmentation of PSD across 56 children with ASD.

Q8: Lines 352-354: “The U-Net backbone based architecture was employed given the constrained training dataset, as implemented in previous PSD segmentation studies 69. “

There is no reference #69 provided in this paper. The authors should recheck their references for accuracy.

Response: We have revised all references.

REVIEWERS' COMMENTS:

Reviewer #2 (Remarks to the Author):

The authors have done a satisfactory job at addressing my previous comments and suggestions, thank you. No further suggestions on my end. I look forward to seeing this interesting study published!

Reviewer #4 (Remarks to the Author):

The authors have addressed all of my concerns and provided clarification and supplementary information. Their overall efforts have improved the quality of the manuscript for publication.